# Dandelion (*Taraxacum officinale* L.) as a Source of Biologically Active Compounds Supporting the Therapy of Co-Existing Diseases in Metabolic Syndrome

**DOI:** 10.3390/foods11182858

**Published:** 2022-09-15

**Authors:** Małgorzata Kania-Dobrowolska, Justyna Baraniak

**Affiliations:** Department of Pharmacology and Phytochemistry, Institute of Natural Fibres and Medicinal Plants, National Research Institute, Wojska Polskiego 71b Str., 60-630 Poznan, Poland

**Keywords:** dandelion, metabolic syndrome, antioxidant activity, hypolipidemic effect, anti-diabetes, antiplatelet activity

## Abstract

Nowadays, many people are struggling with obesity, type 2 diabetes, and atherosclerosis, which are called the scourge of the 21st century. These illnesses coexist in metabolic syndrome, which is not a separate disease entity because it includes several clinical conditions such as central (abdominal) obesity, elevated blood pressure, and disorders of carbohydrate and fat metabolism. Lifestyle is considered to have an impact on the development of metabolic syndrome. An unbalanced diet, the lack of sufficient physical activity, and genetic factors result in the development of type 2 diabetes and atherosclerosis, which significantly increase the risk of cardiovascular complications. The treatment of metabolic syndrome is aimed primarily at reducing the risk of the development of coexisting diseases, and the appropriate diet is the key factor in the treatment. Plant raw materials containing compounds that regulate lipid and carbohydrate metabolism in the human body are investigated. Dandelion (*Taraxacum officinale* F.H. Wigg.) is a plant, the consumption of which affects the regulation of lipid and sugar metabolism. The growth of this plant is widely spread in Eurasia, both Americas, Africa, New Zealand, and Australia. The use and potential of this plant that is easily accessible in the world in contributing to the treatment of type 2 diabetes and atherosclerosis have been proved by many studies.

## 1. Introduction

Metabolic syndrome is affecting an increasing number of people in almost all well-developed countries. The number of people, including children and young adults, with type 2 diabetes, obesity, and atherosclerosis is growing every year. These people often do not receive comprehensive treatment, and each diagnosed component disorder of the metabolic syndrome is treated separately. At the same time, there is a growing awareness in society of the importance of proper diet and physical activity in the prevention and treatment of many diseases. People with coexisting diseases classified as metabolic syndrome, according to the definition, in addition to traditional treatment, are increasingly reaching for other methods, such as the use of herbal medicine. Herbs have accompanied man for years in medicine, cosmetology, as well as in the kitchen. There are a number of plant raw materials that affect lipid and carbohydrate metabolism and improve digestion. One such plant is the dandelion. Dandelion is used both as a medicinal agent and as food. The root is a substitute for cereal coffee, the leaves are eaten raw in salads, and syrups are made from the flowers. Dandelion has many chemical compounds that affect lipid metabolism, protect the liver, regulate blood sugar, and affect digestion and, indirectly, obesity. In addition, some compounds in dandelion regulate platelet aggregation and affect blood pressure regulation. It seems that all these properties recommend this plant for use in complementary therapy in the treatment of coexisting diseases in metabolic syndrome.

## 2. Definition and Etiology of Metabolic Syndrome (MetS)

Metabolic syndrome is the co-occurrence of factors such as central (abdominal) obesity, elevated blood pressure, and disorders of sugar and fat metabolism in the human body, which eventually leads to the development of cardiovascular disease and type 2 diabetes. Lifestyle has an impact on the occurrence of metabolic syndrome. The treatment of metabolic syndrome is primarily aimed at reducing the risk of developing diabetes, hypertension, and cardiovascular disease.

The first definition of metabolic syndrome was proposed by the World Health Organization (WHO) in 1998 [1]. The components of metabolic syndrome include type 2 diabetes, insulin resistance, abnormal glucose tolerance or abnormal fasting glucose, and at least two of the other criteria: microalbuminuria, reduced HDL (high-density lipoproteins cholesterol) or elevated triglycerides, or European Group for the Study of Insulin Resistance (EGIR); microalbuminuria was not considered as a component of the metabolic syndrome. Insulin resistance/hyperinsulinemia occurring together with a fasting blood glucose above or equal to 110 mg/dL (IFG), or impaired glucose tolerance (IGT), the presence of hypertension; elevated triglycerides and/or reduced HDL levels; and abdominal obesity were considered the main criteria for the diagnosis of MetS. Abdominal obesity was assessed by waist measurement rather than waist/hip ratio (WHR) or body mass index (BMI), as per the WHO definition [2]. Subsequent modifications of the MetS definition reduced the emphasis on the relevance of a specific criterion occurrence and focused on the simultaneous occurrence of at least three of the above-mentioned criteria. The focus was on simplifying the diagnosis of MetS in clinical practice by concentrating on identifying people who have an increased risk of cardiovascular disease and treating lipid and non-lipid risk factors, with particular attention paid to insulin resistance. As a result of this approach, there have been changes introduced in the diagnosis of MetS included in the report National Cholesterol Education Program, Adult Treatment Panel III (NCEP-ATP III). It was found that the MetS diagnosis criteria did not require the determination of insulin resistance, making it simple in clinical practice. Although central obesity was recognized as a risk factor underlying the development of metabolic syndrome, all components of MetS were treated equally [3]. The criteria developed by the International Diabetes Federation (IDF-International Diabetes Federation) focus on the co-occurrence of abdominal obesity (waist circumference) together with at least two factors, such as elevated triglycerides, reduced HDL-cholesterol fraction, elevated blood pressure, elevated fasting glucose, or diagnosed type 2 diabetes. As with the NCEP-ATPIII criteria, the authors concluded that determining insulin resistance, which is not easy to measure, is not a requirement for diagnosing MetS. In the report, the authors noted that abdominal obesity is strongly associated with insulin resistance, and measuring abdominal circumference is easy and quick [4,5,6]. According to the guidelines of the Polish Forum for the Prevention of Cardiovascular Diseases (PFPChUK), updated in 2015, metabolic syndrome is a clinical condition characterized by the co-occurrence of multiple interrelated metabolic factors that increase the risk of developing atherosclerotic cardiovascular disease and type 2 diabetes [7]. According to Polish studies, metabolic syndrome includes abdominal obesity and impaired glucose tolerance (insulin resistance and/or hyperinsulinemia), dyslipidemia (high triglycerides and/or low HDL fraction cholesterol), as well as hypertension, and the activation of pro-inflammatory and pro-thrombotic processes. The criteria for the diagnosis of metabolic syndrome according to the PFPChUK are as follows: an increased waist circumference equal to or greater than 80 cm in women and equal to or greater than 94 cm in men, a triglyceride level equal to or greater than 150 mg/dL (1.7 mmol/L) or the use of medications to reduce it, a fasting glucose level equal to or greater than 100 mg/dL (5, 6 mmol/L) or the use of hypoglycemic drugs, and reduced HDL cholesterol less than 50 mg/dL (1.3 mmol/L) in women and less than 40 mg/dL (1.0 mmol/L) in men or the use of drugs to increase its concentration, elevated systolic blood pressure equal to or higher than 130 mm Hg and/or diastolic blood pressure equal to or higher than 85 mm Hg or the use of hypotensive drugs in patients with a positive history of hypertension. According to the team’s study of the PFPChUK, if a patient meets at least three of the above criteria, metabolic syndrome can be diagnosed. The reasons for the development of MetS are linked to several complex mechanisms that have yet to be fully elucidated. There is some debate as to whether the individual elements of the MetS form separate pathological states or whether they are subject to a common broader pathogenetic process. In addition to genetic and epigenetic factors, some lifestyle and environmental factors such as overeating and physical inactivity have been identified as major contributors to the development of MetS. Disorders such as atherosclerosis, type 2 diabetes, and obesity are classified as diet-related diseases. A poor diet (consuming too many calories) is believed to increase the risk of visceral adipose tissue accumulation. One hypothesis for the development of MetS considers visceral obesity as an activating factor in insulin resistance, chronic inflammation, and neurohormonal activation [7]. Metabolic syndrome factors are presented in Figure 1.

## 3. Lifestyle vs. Metabolic Syndrome

Lifestyle demonstrates a huge impact on the development of metabolic syndrome. Its change can significantly reduce the risk of MetS’ possible occurrence. The results from the Diabetes Prevention Program Research Group indicated that lifestyle modification by reducing body weight by 7% and increasing physical activity to 150 min per week reduced the incidence of diabetes more effectively than metformin treatment in people without diabetes but with prediabetes [8,9,10,11,12]. Therefore, one of the first steps to be taken in treating metabolic syndrome is to change the lifestyle by reducing the caloric content of the meals, increasing physical activity, and reducing body weight. Health problems and conditions such as diabetes, obesity, and hypertension are treated with approved drugs providing clinically proven effects and safety of use. However, it also appears that many natural plant materials can be helpful in maintaining normal blood glucose and cholesterol levels. Among medicinal plants that may be efficient in the prevention of type 2 diabetes mellitus, the most popular are: galega (*Galega officinalis* L.), common bean (*Phaseolus vulgaris* L.), fenugreek (*Trigonella foenum-graecum* L.), alfalfa (*Medicago sativa* L.), white mulberry (*Morus alba* L.), ginger (*Zingiber officinale* Rosc.), maize (*Zea mays* L) [13]. Among the plants with the ability to affect lipid metabolism are: garlic (*Allium sativum* L.), turmeric (*Curcuma longa* L.), milk thistle (*Silybum marianum* L.), cardoon (*Cynara cardunculus* L.), Panax ginseng (*Panax ginseng* C.A. Meyer). One of these herbal plants with beneficial pharmacological effects on the set of disease factors included in the metabolic syndrome is dandelion (*Taraxacum officinale* F.H. Wigg.). 

## 4. Dandelion—Plant Characteristics 

Dandelion (*Taraxacum officinale* L. syn. *Taraxacum vulgare* L.), belonging to the Asteraceae family, is a pharmacopeial, edible plant. It probably originated from Europe; it also gradually spread to Asia, then North America, and later to some South American countries. In many European countries, it is a common weed growing in fallow fields, roadsides, meadows, and lawns. Dandelion is a perennial weed with sturdy taproot, long green leaves organized in a rose-like manner, single yellow flowers, and characteristic cotton-like fruits with many seeds that are scattered by the wind [14]. The pharmacopeial raw materials are the roots of the dandelion (Taraxaci radix), herba, and also flowers. The traditional uses of dandelion that are mentioned in the literature concern its use as a remedy in kidney diseases, diabetes, bacterial infections, diuretic, liver, kidney, and spleen disorders, and as an anti-inflammatory factor [15]. On the other hand, dandelion parts are used as food, mainly as a salad ingredient, young leaves are placed in many dishes, and the inulin-rich roots are used as substitutes for coffee or tea [15]. It has been detected that approximately 100 g of fresh leaves contain 88.5 g of water, 19.1 g of crude protein, 6.03 g of crude fat, 10.8 g of crude fiber, and 0.67 g/100 g dry matter of calcium, 6.51 g/100 g dry matter of potassium, 3.99 g/100 g dry matter of zinc, 12.6 mg/100 g dry matter of tocopherols, 156.6 mg/100 g dry matter of L -ascorbic acid and 93.9 mg/100 g dry matter of carotenoids [16]. Dandelion flower extracts can be used as flavor additives in many food products, such as desserts, candies, baked cakes, puddings, and other similar food products [17]. The main active compounds of dandelion are presented in Figure 2.

Dandelion roots contain mainly sesquiterpene lactones and triterpenes and sterols (taraxasterol, taraxerol, cycloartenol, beta-sitosterol, stigmasterol) [18]. Lactones have a bitter taste and are often an ingredient in products that stimulate digestion. The literature evidence suggests that phenolic acids and sesquiterpene lactones are the main components of the dandelion root responsible for its antidiabetic potential [19]. Dandelion leaves and flowers contain polyphenols, mainly hydroxycinnamic acid derivatives (HCAs) and flavonoids (apigenin and luteolin derivatives) [20,21,22]. They are characterized by strong antioxidant and hypocholesterolemic properties. HCAs induce antiradical and protective effects against oxidative processes [22], while flavonoids inhibit the formation of reactive oxygen species and nitrogen by inhibiting NO synthase and COX-2 protein expression [23,24]. Chicoric acid is effective in preventing the formation and worsening of the atherosclerosis process [24]. Dandelion roots also contain significant amounts of inulin [14,20]. Inulin is a naturally occurring polysaccharide belonging to a class of dietary fibers known as fructans. This plant is also an important source of vitamins (A, C, E, K, and B) and minerals (for example, iron and silicium), sodium, copper, zinc, magnesium, and manganese) [14,25]. Dandelion leaves are also a rich source of potassium, which may be related to the plant’s diuretic activity [26]. Selected phytochemicals of dandelion are presented in Table 1.

## 5. Pharmacological Activity of Dandelion for Potential Use in the Treatment of Metabolic Syndrome (MetS) 

### 5.1. Antidiabetic Effect

According to the data provided in the scientific journals, plant products and plant-derived compounds exhibit antidiabetic effects through mechanisms such as reducing the activity of enzymes (α-amylase with β-galactosidase and α-glucosidase) that break down sugars, including polysaccharides, inhibiting renal glucose reabsorption and flow through potassium channels [19]. Different extracts (methanolic, chloroform, aqueous, petroleum ester) of dandelion root were tested for antidiabetic activity in mice with normal glycemia and alloxan-induced diabetes. In addition, the authors carried out in vitro glucose uptake assays using HepG2 and 2-NDBG. The results from the in vivo study showed that an aqueous extract of Taraxacum officinale root (400 mg/kg) caused a significant decrease in blood glucose levels (62.33%, *p* ≤ 0.05), while other extracts (*p* > 0.05) showed a statistically insignificant activity in mice with alloxan-induced diabetes. No effect of the extracts on glycemia was noted in non-diabetic mice. The extracts lowered glucose levels (*p* > 0.05) in the subcutaneous glucose tolerance test. The aqueous extract showed significantly higher glucose uptake (149.6724%, *p* ≤ 0.05). A phytochemical examination of the aqueous extract confirmed a higher total phenolic content than flavonoids, and chlorogenic acid, protocatechuic acid, and luteolin-7-glucoside were identified [40]. Dandelion extract also inhibited the formation of advanced glycation end products (AGEs) (IC50 = 69.4 mg/L) more effectively than the drug commonly used in diabetes, named aminoguanidine (IC50 = 138 mg/L) [41]. Dandelion leaf and root extracts and taraxinic acid β-d-glucopyranosyl ester activated nuclear erythroid-associated transcription factor 2 (Nrf2) in human hepatocytes. The leaves of *Taraxacum officinale* induced the Nrf2 target gene Hmox1. Taraxinic acid β-d-glucopyranosyl ester isolated from the leaves was found to increase Nrf2 transactivation in a dose-dependent manner. The results obtained by Esatbeyoglu et al. (2017) [34] suggest that the antioxidative activity of dandelion leaf extract is responsible for the taraxinic acid β-d-glucopyranosyl ester [34]. Other studies have tested the effect of dandelion extract on insulin secretagogue activity. Dry ethanolic extracts of *Taraxacum officinale* at concentrations ranging from 1 to 40 µg/mL were tested in vitro for insulin release from INS-1 cells in the presence of 5.5 mM glucose, with glibenclamide as a control. Insulin secretagogue activity could be observed for dandelion extracts at a concentration of 40 µg/mL [42]. Alpha-glucosidase was also inhibited by aqueous extracts of *Taraxacum officinale* depending on the origin (alpha-glucosidase from baker’s yeast, rabbit liver, and rabbit small intestine) [43,44] demonstrated the antihyperglycemic effect of a herbal preparation containing 9.7% Taraxaci radix (*Taraxacum officinale* F.H. Wigg.) in an experiment on mice with alloxan-induced non-obesity diabetes (NOD). It was noted that the extract statistically significantly reduced glucose and fructosamine levels. [44]. A follow-up study revealed the effect of the dandelion extract on the catalytic activity of glutathione S-transferases (GSTs) and the formation of malondialdehyde (MDA) in the liver of mice as an indicator of oxidative stress in early diabetes. After a 7-day administration of the dandelion extract (at a dose of 20 mg/kg body weight) to NOD diabetic mice, a significant increase in catalytic GST concentration and a statistically insignificant decrease in MDA concentration were observed [44]. Similar results were obtained by Cho et al. 2002 [45] after administering an aqueous extract of dandelion leaves to rats with streptozotocin-induced diabetes. They observed a decrease in MDA levels in the rats’ liver and a significant reduction in serum glucose levels [45].

α-Glucosidase is an enzyme responsible for breaking down complex carbohydrates: di-, oligo-, and polysaccharides into simple sugars, including glucose. By inhibiting the decomposition of the alpha bonds of carbohydrates, inhibitors of α-glucosidase reduce the absorption of glucose into the blood from the gastrointestinal tract, resulting in lower postprandial glycemia. It was found that the dandelion extracts showed the ability to inhibit alpha-glucosidase [46]. In vitro studies conducted by Mir et al. (2015) [47] on methanolic and aqueous extracts of dandelion leaves, roots, and flowers confirmed their potential to inhibit α-amylase and α-glucosidase activity. It was noted that aqueous extracts inhibited enzymes more strongly than methanolic extracts, with leaf extracts showing the highest activity, followed by root extracts and the weakest activity from dandelion flowers [47]. Li and his team, based on the obtained experimental results, concluded that the aqueous extract of dandelion root, with a composition of polysaccharides (63.92 ± 1.82 mg/g), total flavonoids (2.57 ± 0.06 mg/g), total phenolic compounds (8.93 ± 0.34 mg/g), and saponins (0.54 ± 0.05 mg/g) statistically showed a significant ability to inhibit α-glucosidase and α-amylase activities [48]. Synergism was also observed between the effects of dandelion root extract and Astragalus (*Astragalus* L.) extract. In addition, it was found that mixing the extracts from these plants could alleviate insulin resistance in IR-HepG2 cells.

Choi and his team in 2018 [49] isolated from *Taraxacum officinale*, in addition to the 22 previously known compounds listed below (no. 1–22), three new butyrolactones (1–3) and three butanates (4–6), or taraxioside A-F (Figure 3) (1) 1,2,5-tri-O-p-hydroxyphenylacetyl-L-chiro-inositol, (2) chrysoeriol, (3) 5,7,30-hydroxy-40,50-dimethoxy flavone, (4) methyl 3,4-dihydroxycinnamate, (5) 5,7,40-hydroxy-30,50-dimethoxy flavone, (6) luteolin, (7) 3-glycerindole, (8) calquiqueignan D, (9) calquiquelignan E, (10) tricin 40-O-[threo-b-guaiacyl-(700-Omethyl)-glyceryl] ether, (11) tricin 40-O-[erythro-b-guaiacyl-(700-O-methyl)-glyceryl] ether, (12) loliolide, (13) epiloliolide, (14) annuionone, (15) 11b,13-dihydrotaraxinic acid b-O-glucopyranoside, (16) 3,4-dihydroxy-5,7-megastigmadien-9-one, (17) komaroveside A, (18) 6S,9R-roseoside, (19) 6S,9S-roseoside, (20) adenosine, (21) aesculetin-7-O-b-D-glucopyranoside, and (22) syringin. Their chemical structures were determined by interpreting the spectroscopic data and comparing them with data from the literature. The authors evaluated all isolates for their α-glucosidase inhibitory activity. New compounds I through VI (IC50 145.3–181.3 μM) showed inhibitory activity similar to acarbose (IC50 179.9 μM). Compounds 1 and 6 were the strongest inhibitors, with IC50 values of 61.2 and 39.8 μM, respectively. Compounds II and 6 showed mixed-type inhibition, while compound 1 and acarbose showed competitive inhibition [49].

Perumal et al. 2022 [50] studied the antidiabetic potential of a combination of dandelion and *Momordica charantia* extracts. They determined antidiabetic properties in vitro; the inhibition of α-amylase, α-glucosidase, and dipeptidyl peptidase-4 (DPP-4), and glucose-uptake in L6 muscle cells.

The authors of the study concluded that the antidiabetic efficacy of the combination of the tested herbs was better than the aforementioned herbs used alone. A glucose tolerance test in a study involving rats with streptozotocin–nicotinamide (STZ-NA)-induced diabetes proved that the combination of herbs tested lowered blood glucose levels comparable to the effects of glibenclamide and metformin. The combination of herb extracts showed better antidiabetic properties; it increased the activity of DPP-4, α-amylase, and α-glucosidase [50]. Therefore, the authors suggest that combinations of herbs have a better phytotherapeutic potential for treating type 2 diabetes. In another study conducted by Cho et al. 2002 [45] on a rat model of streptozocin-induced diabetes, the administration of 2.4 g of dandelion aqueous extract/kg of diet reduced postprandial blood glucose levels [45]. In another study conducted in 2012 by Nnamdi and his team [51], the effects of dandelion leaves and roots on streptozotocin (STZ)-induced diabetes in rats were tested. The results provided evidence of a hypoglycemic effect after twice-daily administration of an aqueous or alcoholic extract of *Taraxacum officinale* leaves and roots in amounts of 300 and 500 mg extract/kg b.w. [51]. It was found that the ethanol extract can increase carbohydrate metabolism. It was also suggested that the ethanol extract is more effective than the aqueous extract and that the roots are more therapeutically effective than the leaves in the treatment of diabetes. The experimental results also showed that the effects of the *Taraxacum officinale* extracts were dose-dependent. However, the authors did not analyze the composition of the extracts, so it is not possible to conclude which components of the dandelion are responsible for such activity [51]. In vivo studies in rat models of non-alcoholic steatohepatitis (NAFLD) treated with dandelion extracts showed significant reductions in hepatic lipid accumulation, liver tissue and body weight, and serum cholesterol levels. After the administration of dandelion leaf extracts, insulin resistance was found to be reduced through activation of the AMPK (5′ adenosine monophosphate-activated protein kinase) pathway. Dandelion products, due to the presence of polyphenols and flavonoids in their composition, can regulate the expression of several genes whose dysfunctions contribute to lipid deposition, oxidative stress, and insulin resistance [52]. With the increasing evidence that non-alcoholic fatty liver disease increases the risk of developing type 2 diabetes, it is assumed that non-alcoholic fatty liver disease and non-alcoholic steatohepatitis are specific clinical manifestations of type 2 diabetes through the coexisting process of lipid deposition, chronic inflammation, and liver fibrosis [53]. Other studies have confirmed that polyphenols in dandelion leaf and stem extracts are useful in the treatment of type 2 diabetes and obesity. An ethanolic extract of dandelion was proved to inhibit the formation of advanced glycation end products at IC50 = 69.4 mg/L compared to the antiglycation drug-aminoguanidine (IC50 = 138 mg/L) [54]. Dandelion components also show activity in regulating the pathways responsible for insulin release, most likely by inhibiting certain enzymes involved directly and indirectly in carbohydrate breakdown in the Krebs cycle and glycolytic cycle. The mechanism of insulin release in β cells is a complex process. The ethanolic extract of dandelion at a concentration of 40 μg/mL significantly increased insulin secretion in in vitro studies on the rat INS-1 cell line [42]. In a study performed by Tousch et al. (2008) [32], it was noted that chlorogenic acid CGA is an inhibitor of glucose-6-phosphatase (G6P) in the rat liver and may contribute to intensifying glucose transport, thereby increasing ATP production and stimulating insulin secretion [32]. It is thought that CGA may also regulate β-cell function [55]. An in vivo experiment showed that CGA significantly increases hepatic mRNA expression by interacting with peroxisome proliferator-activated receptor alpha (PPAR-α). Thus, it is investigated that CGA may contribute, through the activation of peroxisome proliferator-activated receptor alpha (PPAR-α) and the stimulation of glucagon-like peptide GLP-1 production, to restore β-cell function, and thus aid in the treatment of type 2 diabetes [56]. 

### 5.2. Impact on Lipid Profile

To determine the possible use of dandelion preparations as a natural anti-obesity agent, Zhang et al. (2008) [57] examined its inhibitory activity against pancreatic lipase in vitro and in vivo. The inhibitory activity of a 95% ethanol extract of *T. officinale* and Orlistat was measured using 4-methylumbelliferyl oleate (4-MU oleate) as a substrate at concentrations of 250, 125, 100, 25, 12.5, and 4 µg/mL. To determine pancreatic lipase inhibitory activity in vivo, mice (*n* = 16) were orally administered corn oil emulsion (5 mL/kg) alone or with 95% ethanolic extract of *T. officinale* (400 mg/kg). The plasma triglyceride levels were measured at 0, 90, 180, and 240 min after administration. It was found that the 95% ethanol extract of *T. officinale* and Orlistat inhibited porcine pancreatic lipase activity by 86.3% and 95.7% at a concentration of 250 µg/mL. The *T. officinale* extract showed a dose-dependent inhibition of IC(50) = 78.2 µg/mL. In addition, it was discovered that a single oral dose of the extract inhibited the increase in plasma triglycerides at 90 and 180 min (*p* < 0.05) [57]. Other in vitro studies have confirmed that flavonoids related to quercetin and luteolin from dandelion inhibit porcine pancreatic lipase [58]. Mice (C57BL/6) that were fed dandelion leaf extract and a high-fat diet had lower serum triglycerides and total cholesterol compared to the control group [59]. Similar findings were reported for rabbits fed a high-cholesterol diet (1% cholesterol) with dandelion root or leaves for four weeks. In these animals, it was also observed that the level of serum triglycerides was significantly lower compared to the control group [60]. Similar issues were the subject of another research paper. The investigation proved that dandelion extracts had an inhibitory effect on adipocyte differentiation and lipogenesis activity in 3T3-L1 pre-adipocytes. The HPLC analysis of the three plant extracts obtained from leaves and roots, which were used in this study, as well as a commercial root powder, showed the presence of caffeic acid and chlorogenic acid as the main phenolic component. It was found that there was no cytotoxicity effect in the concentrations of the extracts used in the experiments—MTT test. The authors inferred that dandelion extracts could affect adipogenesis and lipid metabolism [61]. Other researchers have searched for other biologically active compounds from a number of plants that would show properties that reduce triglyceride accumulation and increase lipolysis and induce investigated apoptosis. The work demonstrated that 18–22% *v*/*v* aqueous-ethanol extract of dandelion root effectively induced the apoptosis of human primary visceral pre-adipocytes during their differentiation while enhancing lipolysis [62]. An interesting study was conducted to reveal how the addition of dandelion extract at a rate of 0.8% to the carp’s daily feed ration would affect the hydrochemical parameters (pH, dissolved oxygen, and electrical conductivity). It was found that carp fed a diet supplemented with dandelion extract did not show improved fish production characteristics compared to those found for carp from the control group. Instead, carp from the experimental groups had higher survival rates, final weights, average individual weight gain, and specific growth rates (SGR), but the differences were not statistically significant. Feed supplementation with dandelion extract significantly reduced plasma cholesterol (by 4.76%) and triglycerides (by 61.2%), which may be an interesting finding for breeders (*p* ≤ 0.05) [63]. 

### 5.3. Impact on Blood Pressure

Oxidative stress is one of the factors co-responsible for the development of hypertension. In subsequent in vitro and in vivo studies, it was verified whether dandelion leaf and root extracts have sufficient antioxidant potential that is able to influence the reduction in hypertension-inducing factors. In this study, the malondialdehyde (MDA) levels were determined in lipid peroxidation assays and in rats with hypertension stimulated by free radical production. Oxidative stress was induced by N ω-nitro- L-arginine methyl ester. Aremu et al. (2019) [64] noted that the extract increased antioxidant activity and reduced lipid peroxidation in the heart, liver, kidney, and brain of the tested rats. The authors suggest that the phenolic compounds present in the extracts may also regulate nitric oxide synthase (NOS) levels and activity by affecting kinase signaling pathways and intracellular Ca^2+^ associated with NOS phosphorylation and NO production. In addition, phenolic compounds can also affect the inhibition of endothelin-1 (vasoconstrictor) and endothelial NADPH oxidase, but more research is required to confirm these theses [64]. 

### 5.4. Effects on Blood Coagulation

The inhibitory effect on platelet aggregation in humans by ethanolic extracts of dandelion root (*Taraxacum officinale* F.H. Wigg.) was examined. The extracts showed dose-dependent inhibition of platelet aggregation, where the maximum inhibition was 85% at a concentration equivalent to 0.04 g dried root/mL human platelet-rich plasma (PRP). The effect was obtained regardless of whether platelet aggregation was induced by arachidonic acid or collagen. The extracts were fractionated into two groups of compounds with masses above (Mr > 10,000) and below (Mr < 10,000). The fraction containing low-molecular-weight polysaccharides (Mr< 10,000) resulted in 91% inhibition, while the other fraction enriched in triterpenes and steroids (Mr > 10,000) showed 80% inhibition of platelet aggregation. Both at a concentration equivalent to 0.04 g raw material/mL PRP [65]. In another in vitro study, Lis and the team (2018) [66] determined the antiplatelet and antioxidant properties of four standardized phenolic fractions of dandelion. The following fractions were analyzed: two leaf fractions of 50% and 85% methanol and two petal fractions of 50% and 85% methanol. The hemostatic activity in the plasma was determined: activated partial thromboplastin time (APTT), prothrombin time (PT), and thrombin time (TT). It was found that none of the dandelion fractions tested caused damage to human platelets over the entire range tested. The results of the study show that dandelion, especially its aboveground parts containing hydroxycinnamic acid, which has antioxidant and antithrombotic effects of the hemostatic system; that is, they may be promising preparations in the prevention of cardiovascular diseases, especially those related to changes in hemostasis and oxidative stress [66]. Studies concerning the antioxidant potential of dandelion preparations have confirmed that a diet rich in dandelion preparations can be helpful in treating diseases related to oxidative stress and hemostatic disorders. Although a great number of chemical compounds present in dandelion, such as hydroxycinnamic acid and sesquiterpene lactones, were previously detected, new compounds are still being discovered and analyzed. Recently, inositol 4-hydroxyphenylacetate (PIE) esters have been characterized. In work performed by Jedrejek et al. 2019 [67], five fractions of dandelion extract were analyzed, where each was characterized by different contents of active compounds. Detailed LC-MS and chemical tests of the dandelion fractions identified about 100 phytochemical compounds, including new ones. In all concentration ranges tested (0.5–50 μg/mL), the dandelion root preparations did not cause platelet hemolysis. The results indicate that dandelion roots constitute a safe and readily available source of different classes of natural compounds with antioxidant, anticoagulant, and antiplatelet effects [67].

The aim of another in vitro study was to evaluate the activity of dandelion extracts, which were standardized for chicoric acid content. Four phenolic fractions extracted from leaves (fractions A and B) and flower petals (fractions C and D) were characterized by different concentrations of chicoric acid. The biomarkers of oxidative stress, coagulation, and platelet activation parameters were determined. The results suggest that chicoric acid has antioxidant and anti-adhesive potential. The authors noted that the fraction richest in chicoric acid (leaf fraction A) possesses anti-adhesive and anti-aggregation properties stronger than chicoric acid alone. These findings strongly suggest the possibility of a synergistic effect between the compounds in fraction A and also the presence of compounds such as phenolic acid derivatives and flavonoids, which may exhibit stronger properties than chicoric acid [68]. Relying on previous in vitro studies, another research group decided to test the effects of the same extracts in in vivo studies, conducting tests on rats. The animals were given a diet enriched in phenolic fractions obtained from dandelion leaves and petals (694 mg/kg diet/day) for 4 weeks. The phenolic fractions obtained from the dandelion leaves and petals contained, respectively, 4.10 ± 0.05 and 1.41 ± 0.07 mg of l-chicoric acid in the daily dose. It was found that supplementation with the petal fraction increased plasma thiols. The leaf fraction reduced the level of protein carbonylation and affected the lipid profile—triglycerides, total cholesterol, lipoprotein pooling index, and the plasma atherogenicity index were reduced. The authors concluded that the phenolic fractions from *T. officinale* rich in hydroxycinnamic acids should be considered as potential components of functional foods with beneficial effects on human health [33]. 

The phytochemical analysis of dandelion fruits is also an issue worthy of interest. It should be noted that the root, leaf, and flowers are relatively well studied. Research on dandelion fruits is rarely undertaken. However, fruits are also used in medicine and food. Lis et al. 2020 [37] obtained a methanolic extract of dandelion fruit (E1). Analysis of an extract revealed the presence of hydroxycinnamic acid (HCA) derivatives and flavone derivatives. Several new metabolites were also detected, such as biflavones and some flavonolignans. Multistage fractionation of the methanolic extract of dandelion fruit was carried out. Two fractions were prepared: phenolic acid extract (E2) and flavonoid extract (E3). The E3 extract was divided into four flavonoid fractions: A (luteolin fraction; 880 mg GAE/g), B (philonotisflavone fraction; 516 mg GAE/g), C (flavonolignans fraction; 384 mg GAE/g), and D (flavone aglycones fraction; 632 mg GAE/g). The highest antiradical activity of DPPH was exhibited by fractions A > B > Trolox, medium Trolox > E3 > E2 > E1, and the lowest by C and D. No cytotoxic effect on platelets was noted for any of the dandelion preparations tested. Several different parameters, as well as lipid peroxidation, protein carbonylation, thiol oxidation, and platelet adhesion, were analyzed. The hydroxycinnamic acid extract (E2), flavonoid extract (E3), and luteolin fraction (A) showed the highest antioxidant and antiplatelet potential [37]. 

The antiplatelet potential of four fractions obtained from different parts of the dandelion (fractions A and B from roots; fraction C from leaves; fraction D from petals) on platelet activation and thrombus formation in whole blood were analyzed as well as the effect of the tested fractions on the platelet proteome were also evaluated. The authors found that fraction C from dandelion leaves reduced thrombus formation and platelet activation after collagen stimulation. None of the fractions tested caused changes in the platelet proteome. The preparations obtained from different parts of the dandelion can have a beneficial effect in the prevention and treatment of cardiovascular diseases caused by hyperactivation of platelets [69].

### 5.5. Dandelion vs. Obesity

To determine the possible use of the dandelion preparations as a natural anti-obesity agent, Zhang et al. 2008 [57] measured its inhibitory activity against pancreatic lipase in vitro and in vivo. The inhibitory activity of a 95% ethanol extract of *T. officinale* and Orlistat was measured using 4-methylumbelliferyl oleate (4-MU oleate) as substrate at concentrations of 250, 125, 100, 25, 12.5, and 4 µg/mL. To determine the pancreatic lipase inhibitory activity in vivo, mice (*n* = 16) were orally administered corn oil emulsion (5 mL/kg) alone or with 95% ethanolic extract of T. officinale (400 mg/kg). The plasma triglyceride levels were measured at 0, 90, 180, and 240 min after administration. It was found that 95% ethanol extract of *T. officinale* and Orlistat inhibited porcine pancreatic lipase activity by 86.3% and 95.7% at a concentration of 250 µg/mL. *T. officinale* extract showed a dose-dependent inhibition of IC(50) = 78.2 µg/mL. In addition, it was noted that a single oral dose of the extract inhibited the increase in plasma triglycerides at 90 and 180 min (*p* < 0.05) [57]. The mice were administered dandelion extract, which showed anti-obesity potential through a dose-dependent inhibitory effect on pancreatic lipase activity and an increase in plasma triglyceride levels. The results indicate that *T. officinale* may be an alternative to Orlistat, a drug which often causes adverse effects [57]. In another study, the anti-obesity effects of the dandelion ethanol extract were examined. The ethanolic extract of *Taraxacum officinale* was administered orally at a dose (150 and 300 mg/kg), and Orlistat was used as a reference drug. The experimental rats were fed a high-fat diet. The high-fat diet caused significant increases in body weight, fat mass, serum glucose concentration, as well as cholesterol and triglycerides levels. The authors noted that *Taraxacum officinale* extract significantly reduced body weight, lipid parameters, organ weights, and fat pad mass. The value of the study, however, is diminished by the fact that the content of phytoactive compounds in the extract was not determined. Unfortunately, it is not possible to determine which compounds are responsible for such effects [70]. The possibility of using dandelion extracts to compose formulations and functional foods affecting obesity reduction was also investigated. In the Aabideen et al. 2020 [71] experiment, various aqueous-ethanol extracts were tested for their antioxidant potential. Next, 60% of the extracts with the strongest antioxidant activity were subjected to in vivo testing. BALB/c mice weight gain, fecal fat content, and food intake were compared after an administration of an 8-week fat-rich diet. An increase in the body weight of 44.94% of mice on the high-fat diet (HFD), compared to the NDG control group of 22.21% after eight weeks was observed. After eight weeks on the HFD diet, the obese mice were divided into groups to evaluate the effects of plant extracts on obesity parameters. Treatment with plant extracts and comparative Orlistat was continued for another eight weeks. Mice given the plant extract at 300 mg/kg b.w. reduced their weight to 32.22 ± 1.86 g, and those consuming the drug Orlistat reduced their weight to 30.09 ± 1.61 g compared to the untreated group (HFD) (52.66 ± 2.03 g). The Orlistat-treated mice had a fecal fat content of 11.65%, the 300 mg/kg b.w. extract group had a fecal fat content of 9.92% compared to the HFD mice at 5.67%. Food intake in mice in the HFD + 300 extract groups was 3.65 g/mouse/day, and HFD + Orlistat was 3.8 g/mouse/day compared to mice on the HFD diet of 4.12 g/mouse/day. The findings suggest that dandelion preparations may be an alternative to the use of the drug Orlistat [71]. A study conducted by Majewski and his team detected that the consumption of aqueous dandelion flower syrup (278.2 g/kg diet for four weeks) had beneficial effects on rat blood lipid regulation, which was manifested by increasing HDL fraction, increasing plasma superoxide radical (SOD) scavenging, and decreasing lipid peroxidation. In addition, the liver damage marker ALP content was lowered. The aqueous syrup of dandelion flowers contained hydroxycinnamic acids and flavonoids. Studies have discovered that syrup from *T. officinale* floral water at a dose of 278.2 g/kg of diet affects antioxidant levels and reduces smooth muscle contraction in the blood vessel wall. The authors concluded that phenolic compounds in flower syrup exhibit health-promoting properties. Its antioxidant activity is responsible for these properties [72].

The preparations (extracts, syrups, etc.) obtained from different parts of the dandelion (root, leaves, and flowers) show health-promoting effects. They have the ability to regulate glucose levels and lipid profile, affect digestive enzymes, and indirectly reduce obesity. However, this has been observed mainly in in vitro studies or in animal models. It was stated that there is a need for in-depth studies on this issue with healthy volunteers and people with various cardiovascular diseases, obesity, or type 2 diabetes who would be given dandelion preparations.

## 6. Reports on the Toxic Effects of Dandelion and its Preparation

Dandelion has been consumed as food and used as herbal medicine for centuries, and the side effects of its consumption are rather rare. Dandelion root and dandelion extracts have “generally recognized as safe” status approved by the FDA for use in dietary supplements. Fresh *Taraxacum officinale* leaves and other parts are consumed as food in many countries. 

Many studies on animals have been conducted regarding the potential toxicity of this plant. Toxicological studies have been conducted, and LD_50_ at *per os* administration to mice was determined to be greater than 20 g/kg body weight [73]. In subchronic toxicity studies (4 months), no toxic effect was noted in rats fed with dandelion leaves (33% in the diet) [74]. No acute toxicity was observed in rabbits after the oral administration of dehydrated dandelion plant at a dose of 3–6 g/kg body weight. The LD_50_ (intraperitoneal injection) of the liquid extract of the herb and root for mice was 28.8 g/kg and 36.6 g/kg, respectively. It was only discovered that taraxacum acid esters could cause contact dermatitis [75]. In vitro studies have noted that dandelion infusions can inhibit cytochrome 3A4 (IC50 = 140.6 µg/mL), which may lead to interactions with the metabolism of, for example, immunosuppressive drugs [76]. In studies with rats, it was shown that doses up to 1000 mg/kg b.w. did not cause mortality when administered by the oral route, as did doses of 1600, 2900, and 5000 mg/kg b.w. [64].

Dandelion intake is generally considered safe and well-tolerated in adults if taken in moderation, but some side effects exist, such as diarrhea, upset stomach, or irritated skin. According to Yarnell et al. 2009 [74], it seems that due to its bitter content, dandelion should be consumed with caution by people with diagnosed acute gastroenteritis or reflux esophagitis [74], acute inflammation, or obstruction of the gastrointestinal tract. Allergies to dandelion may also occur. No information on dandelion toxicity or serious adverse effects in humans has been encountered in the scientific literature [77]. 

## 7. Final Remarks

Dandelion is an interesting herbal plant that can be successfully used in the food, pharmaceutical, and cosmetic industries. It manifests multidirectional pharmacological activity that is widely documented in the scientific literature. This paper presents the properties of dandelion, which can be successfully used in the treatment and prevention of metabolic syndrome. A review of available in vivo and in vitro studies indicates that dandelion extracts can prevent diabetic complications, improve lipid metabolism, as well as exhibit inhibitory activity on sugar-degrading enzymes. The aforementioned activity, according to the definition describing metabolic syndrome, is part of the current recommendations for its treatment, the primary aim of which is to prevent the development of diabetes, hypertension, and other cardiovascular diseases. The multidirectional effects of the dandelion and its preparations are presented graphically in Figure 4.

Many questions still remain to be clarified. Therefore, new in-depth scientific research on all biological activities of *Taraxacum officinale* in relation to human health is essential for a thorough understanding of the mechanisms of action of the preparations from this plant.

## Figures and Tables

**Figure 1 foods-11-02858-f001:**
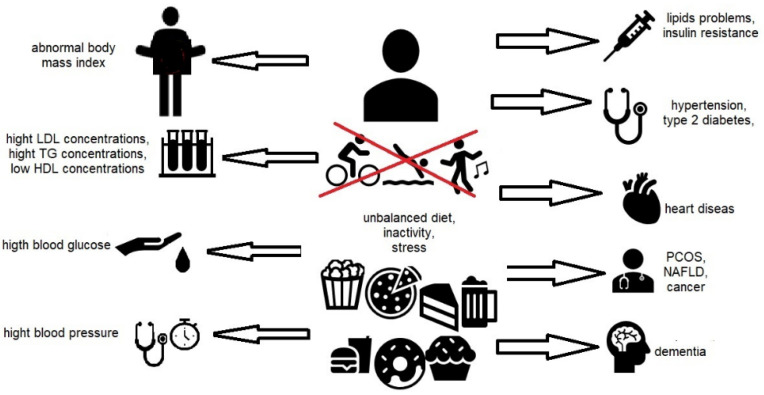
**Metabolic syndrome factors**/PCOS—polycystic ovary syndrome; NAFLD—non-alcoholic fatty liver disease/.

**Figure 2 foods-11-02858-f002:**
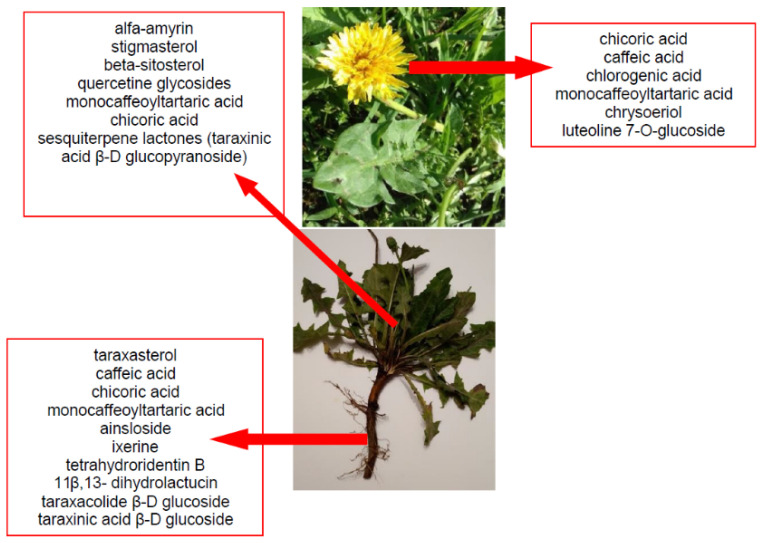
The main active compounds of dandelion.

**Figure 3 foods-11-02858-f003:**
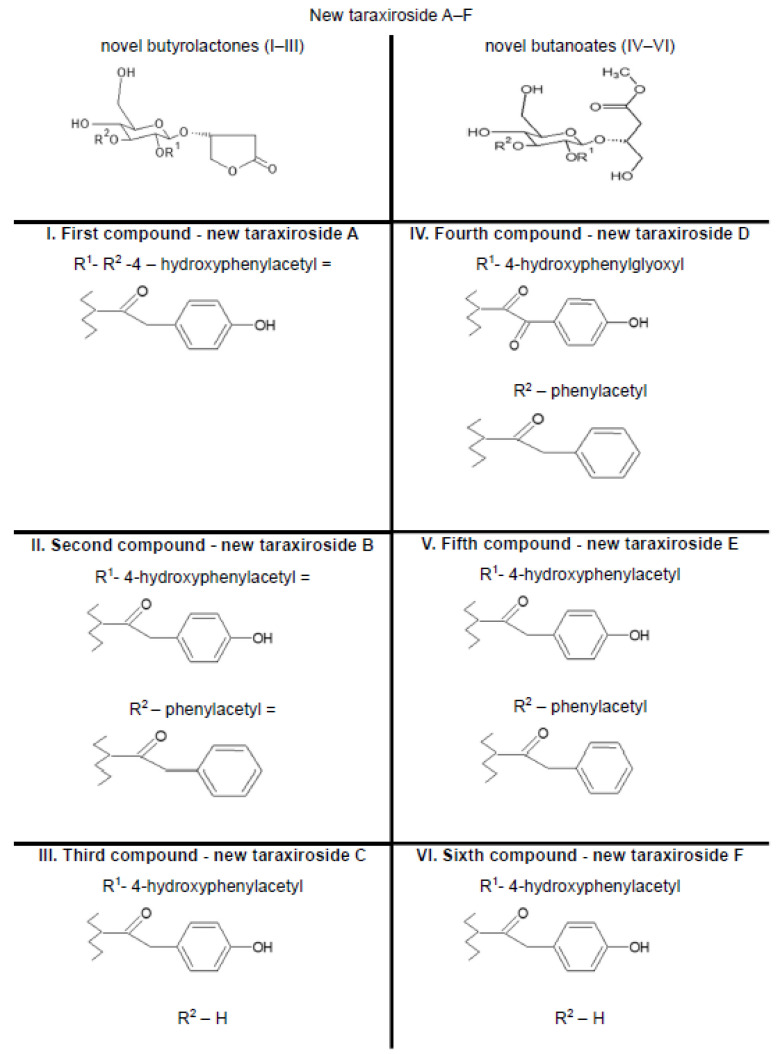
New compounds isolated from *Taraxacum officinale* Choi et al. 2018 [49].

**Figure 4 foods-11-02858-f004:**
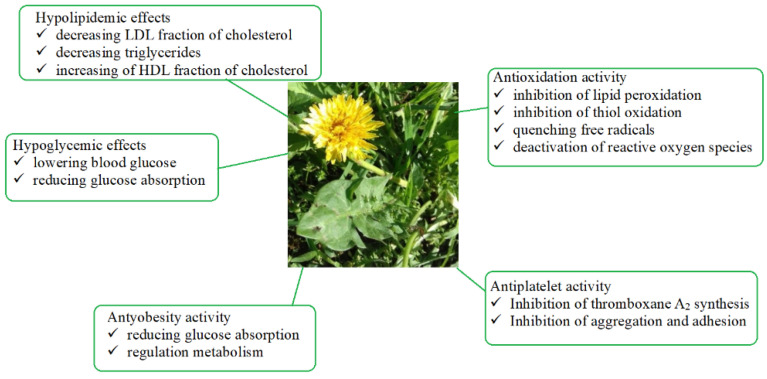
Multidirectional effects of dandelion and its preparations.

**Table 1 foods-11-02858-t001:** Selected phytochemicals of dandelion and their effects.

Name of the Phyto-Component and Parts of the Plant	Structure	Actions	References
taraxasterol (phytosterol) root	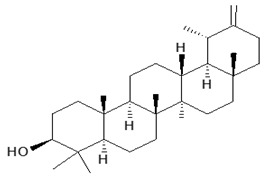 C_30_H_50_O	antihyperglycemic and anti-inflammatory properties	[27]
		anti-inflammatory activity	[28,29]
		decreases protein expression levels of PTBP1 and SIRT1, and may inhibit HBV and be a potential anti-HBV drug,	[30]
stigma sterol (phytosterols) leaf and steam	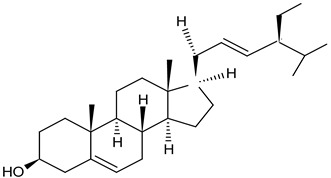 C_29_H_48_O	anti-inflammatory, anti-hyperglycemic, antimicrobial properties	[31]
chicoric acid all parts of the plant	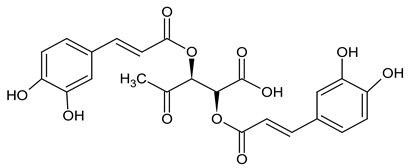 C_22_H_18_O_12_	antidiabetic agent with both insulin-sensitizing and insulin-secreting properties, preventing the formation and/or progression of atherosclerosis, antiradical and protective actions against oxidation processes, meanwhile, flavonoids inhibit the formation of reactive oxygen and/or nitrogen species by suppressing NO synthase and COX-2 protein expression	[32,33]
tetrahydroridentin B sesquiterpen lactone root	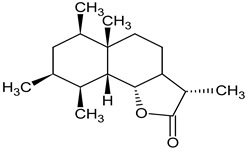 C_15_H_24_O_4_	activated the transcription factor nuclear factor erythroid 2-related factor 2 (Nrf2) in human hepatocytes, induced the Nrf2 target gene heme oxygenase	[34]
chlorogenic acid flower	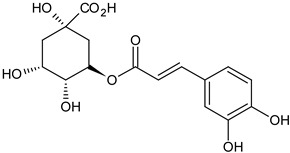 C_16_H_18_O_9_	antioxidant properties	[35]
		anti-inflammatory, antibacterial, antiviral, hypoglycemic, lipid-lowering, anticardiovascular, antimutagenic, anticancer, immunomodulatory	[36]
caffeic acid flower and root	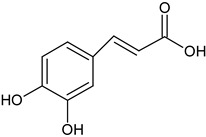 C_9_H_8_O_4_	anti-oxidative and immunostimulatory properties	[35]
hydroxycinnamic acids fruit	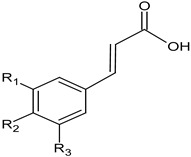 R_2_-OH p-coumaric acid R_3_-OCH_3_; R_2_-OH ferulic acid R_1_-OCH_3_; R_2_-OH; R_3_-OCH_3_ sinapic acid	in experiments on plasma and platelets, using several different parameters (lipid peroxidation, protein carbonylation, oxidation of thiols, and platelet adhesion), the highest antioxidant and antiplatelet potential was demonstrated	[37]
luteolin aboveground plant parts	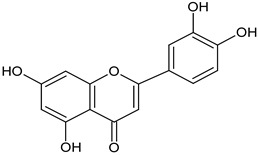 C_15_H_10_O_6_	important role in the amelioration of LPS-induced oxidative stress and inflammation.	[38]
inulin root	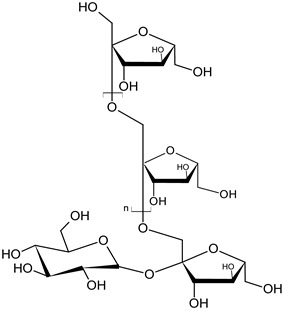 C_6n_H_10n+2_O_5n+1_	influences the development of normal intestinal microflora	[39]

## Data Availability

Not applicable.

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
