# Peer review of "Dandelion (Taraxacum officinale L.) as a Source of Biologically Active Compounds Supporting the Therapy of Co-Existing Diseases in Metabolic Syndrome"

_foods, 2022, doi:10.3390/foods11182858_

Round 1

Reviewer 1 Report

This review summarized the effects of dandelion on metabolic syndrome. However, there are some questions should be resolved.

1. The novelty of this review should be discussed more in “Abstract” and “Final remarks”.

2. In “1. Definition and etiology of metabolic syndrome (MetS)”, I suggested that the authors should simplify the introduction of the history of definition of tMetS and present the current/authoritative definition more clearly. Moreover, in this part, most of description was about the diagnostic criteria rather than etiology of MetS, which is inconsistent with the subtitle.

3. In “Lifestyle vs metabolic syndrome”, why you talk about the herbal medicines? I think that taking Chinese medicine can not be a “lifestyle”.

4. In line 166-167, the author should supplement some related researches.

5. The expression, especially the unit of measurement, in 133-136 confused me. And why you only introduced the components of fresh leaves in detail? The expression should be parallel.

6. In part 4, some parts are focusing on risk factors (4.2 and 4.4), while some are focusing on diseases (4.1, 4.3 and 4.5). The contents are disordered, and you should recompose it.

7. The molecular formula should be listed in Table 1.

8. In line 524-525, the author should supplement some related researches.

9. There are some mistakes in this paper. For example. In line 40, I think EGIR should be “European Group for the Study of Insulin Resistance”. The author should check this paper seriously.

10. The language in this paper should be polished.

Author Response

The response to the review is attached.

Reviewer 2 Report

The topic of this review article is sound. Authors are advised to follow these points:

-          Extensive English editing is needed.

-          Where is the introduction?

-          Summarize the modes of action of Dandelion in an innovative illustrative figure.

-          All scientific names must be italicized (See lines 111-120 and the whole manuscript).

-          L422-486: Take care of paragraphing to make your MS readable.

-          Make use of more papers on this topic like:

Beneficial uses of dandelion herb (Taraxacum officinale) in poultry nutrition, World's Poultry Science Journal, 73:3, 591-602, DOI: 10.1017/S0043933917000459

-          Please correctly follow the guidelines of the journal style.

Author Response

The response to the review is attached.

Reviewer 3 Report

The title of this article is “Dandelion (Taraxacum officinale L.) as a source of biologically active compounds supporting the therapy of co-existing diseases in metabolic syndrome”. This is an interesting topic, and it is an area that needs our attention. However, there are still some areas of the article that need to be revised:

1.      In the "introduction" section of the article, the author can conclude the chapter with a general summary of the content of the article and an outlook on the future development of the field. In addition, the author can present the second paragraph of the section in a segmented manner to better focus the article.

2.      In the "2. Lifestyle vs metabolic syndrome" section of the article, the authors mention the effects of exercise and diet on metabolic syndrome, for which they need to introduce more references.

3.      Figure 2. The main active compounds of dandelion. In this part, the author may add more in-depth discussion and compare their results with the manuscripts recently published in authoritative journals.

4.      The author mentions the effect of dandelion on blood pressure in the article, and this part is very interesting. For this part, the author needs to explore more deeply and give more of his own opinion and vision for the future.

5.      Authors are requested to carefully check the format of the references used in the article to ensure that the references are in the required format.

Author Response

The response to the review is attached.

Author Response

The response to the review is attached.

Round 2

Reviewer 1 Report

This paper could be accepted in present form.